# Technical Complications of Coronary Bifurcation Percutaneous Interventions

**DOI:** 10.3390/jcm11226801

**Published:** 2022-11-17

**Authors:** Gianluca Rigatelli, Marco Zuin, Dobrin Vassilev, Giulio Rodino’, Giuseppe Marchese, Giampaolo Pasquetto

**Affiliations:** 1Interventional Cardiology Unit, Division of Cardiology, Ospedali Riuniti Padova Sud, 35043 Padova, Italy; 2Department of Translational Medicine, University of Ferrara, 44121 Ferrara, Italy; 3Interventional Cardiology Department, MedikaCor Hospital, 7002 Ruse, Bulgaria

**Keywords:** coronary artery disease, coronary bifurcation, percutaneous coronary interventions, complications

## Abstract

Coronary bifurcation percutaneous interventions (PCI) comprise a challenging subset of patients with coronary artery disease. Beyond the well-known debate about single versus double stent strategies, which have different outcomes on mid- and long-term follow up, both strategies may be subject, although rarely, to several different technical complications, rarely reported in clinical trials, which need to be defined, classified, and understood by cardiovascular professionals involved in the management of patients with coronary bifurcation disease. The present paper aims to broaden the knowledge of the range of intraprocedural complications and relative treatment during PCI of coronary bifurcations.

## 1. Introduction

Coronary bifurcation percutaneous interventions (PCI) comprise a challenging subset of patients with coronary artery disease. PCI historically have a worse outcome compared to general revascularization procedures [1,2], but the advent of the third-generation drug-eluting stent (DES) has also drastically improved the results for patients with bifurcation disease. However, coronary bifurcations, per se, remain an independent risk factor for poor PCI outcomes [3]. For many different reasons, the technical complexity of bifurcation disease still represents an issue, with bifurcation PCIs potentially suffering from several intra- and peri-procedural complications, which, although infrequent, still occur. However, such technical problems, which may lead to serious and sometimes life-threatening clinical events, are generally underreported in randomized clinical trials and observational registries. The present paper aims to broaden the knowledge of the range of intraprocedural complications and relative solutions during PCI of coronary artery bifurcation disease, as informed by the current literature and the authors’ experience.

## 2. Data from Current Literature

The complications usually reported in most recent clinical trials include somewhat macrocomplications [4,5,6,7], such as stent thrombosis, side branch (SB) occlusion, acute myocardial infarction (AMI), and cardiac tamponade. However, current literature has hardly focused on the cause underlying such clinical complications, which usually find their cause at a more technical level.

## 3. Basics of Coronary Bifurcation and Treatment in Brief

Beyond the discussion about beneficial effects on clinical outcomes of different strategies to fix coronary artery bifurcation disease, in order to improve the knowledge of technical complications, it is essential to understand the basics of bifurcation and the single versus double stent strategy.

### 3.1. Basics of Coronary Artery Bifurcation

Coronary bifurcation follows the rule of fractal subdivision, known as Murray’s Law [8], which is fundamental to ensure distal coronary perfusion. At any division of the vascular tree, the main vessel (MV) is divided into a main branch (MB) and a SB:Diameter (MV) 3 = Diameter (MB) 3 + Diameter (SB) 3

Coronary bifurcation can be classified using the Medina classification [9], where a three number code indicates absence = 0 or presence of the disease = 1 at MV, MB, or SB, respectively. Then the DEFINITION criteria, derived from the homonymous trial [6], is used to differentiate the bifurcation as simple or complex, based on the length of SB disease, the presence of calcification, multiple lesions, etc. (Figure 1).

### 3.2. Single Strategy

The single strategy, mainly advocated by the European Bifurcation Club (EBC) [10], at least in simple but usually used also in complex bifurcation, is based on the provisional stenting or crossover technique, where the MV is stented across the SB, and the stent diameter is selected upon the distal reference diameter of the MB. The proximal optimization technique (POT), performed with a short balloon dilated at the carina, is then mandatory to correct the malposition in the proximal stent portion and to open the stent struts towards the SB ostium, facilitating a possible further rewiring. With this strategy, the physiological three diameter rule of the bifurcation is maintained [11]. The SB patency is guaranteed by the POT, and in the case that the SB is compromised by the sequence of POT-Side-POT or Re-POT where the SB is rewired, the SB is dilated, followed by another POT to correct the distortion of the stent after SB dilation. In case of severe SB compromise, the provisional technique can be converted to a double stent, implanting a stent at the SB ostium with different techniques (Figure 1).

### 3.3. Double Stent Strategy

Mainly advocated for complex bifurcation disease [12], this interventional strategy is based on the deployment of stents in both the MV and SB using different techniques in which the SB stent can be deployed upfront or after MV–MB stenting. The upfront strategy includes the culotte, where the two stents are deployed sequentially one after the other, resembling women’s lingerie, the double-kissing crush (DK-Crush), where a portion of the SB stent is crushed by the MV stent and two kissing balloon steps are necessary, and the Inverted-T family of techniques, where the SB stent is left protruding 1 mm through a portion of the first ring, as in the Nano-Inverted-T, and the coverage of the SB ostium is partially provided by each of the SB and the MV stents. After MV stenting, the double techniques are T-stenting and T-and-Protrusion (TAP), where the stent on the SB is left protruding into the MV, creating a new and more proximal neocarena (Figure 1). A summary of the different techniques is provided in Table 1.

### 3.4. Technical Complications of Bifurcation Management

Technical complications during coronary bifurcation PCI can be classified based on the level of bifurcation, being that the left main (LM) vessels are obviously different from nonleft main vessels. Secondly, the complications are different in single stent versus double stent procedures, and again, differences can be observed if the complication occurs at the stage of wire crossing, lesion preparation, balloon dilation, stent crossing, stent deployment, wire recrossing, or at the stage of final optimization. A proposed comprehensive classification is shown in Table 2. Relationships of specific complications and different techniques are shown in Table 3.DK: 

### 3.5. Dissection

Dissection of the coronary vessel can occur at any level and stage of the procedure, such as during guiding catheter manipulation, wire crossing, usually distal dissection, during balloon inflation, during plaque modification with atherectomy devices or scoring balloon, and after a stent deployment, which resulted in edge dissection [13]. Independent of the underlying mechanism, flow-limiting dissections after stent deployment are usually treated by stent extension. Occasionally linear dissection at the edge of the stent can be left untreated if non-flow-limiting.

### 3.6. Perforation/Vessel Rupture

Wire perforation occurs usually distally to the target lesion, due to positioning of a hydrophilic wire that is too distal. Prolonged balloon inflation is usually well tolerated and effective without excessive pericardial effusion. If insufficient, coil embolization through a microcatheter is usually effective. A different clinical picture is that of vessel rupture caused by balloon inflation or after stent deployment. The vessel is generally lacerated by an excessive balloon or stent/artery ratio, or because a laminar calcification is pushed out by balloon or stent inflation; this mechanism causes cardiac tamponade very quickly. Rapid prolonged inflation of a new balloon with a 1:1 ratio (Figure 2) with the vessel gives the time to prepare for stent–graft placement or, if not available or not passing, for attempting pericardial drainage to create a pericardial space-venous circuit for reinfusing blood from the pericardium. In extreme cases, cardiac surgery may be the patient’s only chance. Obviously if such a complication occurs during LM bifurcation treatment, it would be particularly catastrophic. It is still possible sometimes to seal the break and complete the intended procedure without complete heparin reversal, if surgery is not available or possible [14].

### 3.7. Wire Entrapment and Fracture

Although very rarely, wires of any type can fracture and break. Any fractured wire, if the portion is short, can be left in the vascular bed once it is covered by a stent [15,16]. A long portion of fractured wire should be removed using a goose-neck catheter of proper dimension (3–6 mm).

### 3.8. Balloon Rupture

Balloon rupture can always happen at any stage of the procedure. This complication is generally due to the presence of coronary calcifications or excessive deployment pressure of the balloon. It can be uneventful, but could also result in vessel rupture, slow flow or no flow if the balloon contains some air, or even cause balloon entrapment if the balloon lacerates irregularly inside a previously deployed stent. Slow or no flow can be treated with air aspiration, intracoronary nitrates or calcium antagonist [17].

### 3.9. Balloon Entrapment

During coronary bifurcation interventions, balloons can remain stuck in highly calcified lesions or be entrapped in a previously malposed stent, or in the attempt to reach the SB in a provisional technique, often when POT has not been properly performed. Maneuvers to disengage the balloon include inflation of a second balloon alongside careful and gentle traction of the balloon shaft, re-expansion of the balloon and slow deflation, and finally forced removal of the whole balloon–wire–catheter unit.

### 3.10. Device Entrapment

A debulking device, such as rotational or orbital atherectomy or general plaque modification tools (scoring balloon or intravascular lithotripsy device) rarely can remain entrapped in a tight lesion similar to what occurs in nonbifurcated lesions. Removal techniques are specific of each device [18].

### 3.11. Stent Loss

With third-generation ultrathin strut stents, stent decrimping and loss is not so infrequent, especially when dealing with complex bifurcation [19], which occurs at a frequency of 0.56% in the general PCI population. To properly maintain the wire inside the lost stent is a different matter compared to having lost the stent and the wire. In the first case, crossing the lost stent with a small 1.25–1.5 balloon and inflating the balloon to try to put the stent in the optimal position (Figure 3) or to place it in a neutral position is the first choice. If the wire is also lost, rewiring and trying to catch it with a balloon is the obvious step. If ineffective, crushing the stent again the wall with another stent can solve the problem. Snaring a lost stent with a coronary snare is always possible, but it is difficult and risky for potentially huge dissections for LM bifurcation. To convert the intended procedure to a DK-crush stenting would be, when possible, a convenient way to solve the problem, depending on the distance between the stent position and the target lesion (Figure 4) [20].

### 3.12. Stent Underexpansion

Underexpansion of a stent usually occurs at high levels of calcification not properly assessed by angiography or intravascular ultrasound/optical coherence tomography, leading to early thrombosis and myocardial infarction. An attempt to inflate a high pressure noncompliant balloon can be made at the risk of vessel rupture. Ad hoc use of intravascular lithotripsy recently has become the most effective method [21].

### 3.13. Abluminal Recrossing

Abluminal recrossing (Figure 5) of a wire during rewiring in the provisional or double stent strategy is frequent and could be not considered a complication, per se. However, this scenario may become a problem, leading to thrombosis and stent deformation if not promptly recognized and treated. This could be a rather frequent issue, especially in malposed stents with ultrathin struts and in such complex two-stent procedures as DK-crush or culotte. These complications may remain unrecognized, especially when occurring in side branch stents, with possible severe outcomes. Therefore, the results should be checked by intracoronary imaging (OCT provides more details than IVUS due to its superior resolution) [11]. Corrective strategies include, in both provisional and dual-stenting techniques, to redo a POT with a larger balloon at a 1:1 ratio with the vessel diameter, to change the wire with a nonhydrophilic one, and finally, to use a microcatheter to address the wire in the center of the stent lumen [11,12].

### 3.14. Stent Deformation

During bifurcation PCI, stent deformation can be caused by too deep of an engagement of the guiding catheter in the proximity of the stent in LM, which deforms and shortens the stent itself. This complication, if not recognized by stent enhancement or IVUS/OCT, can lead to stent thrombosis and myocardial infarction [22,23]. Strategies to overcome this problem are to redo a POT and, if unsuccessful, to implant an additional stent (Figure 6). Stent deformation and shortening can occur also during balloon crossing if the previous stent is malposed.

Another stent deformation complication may occur when removing the balloon, elongating stent struts and inducing a severe malposition (with possible very late stent thrombosis after stopping double antiaggregation).

### 3.15. Stent Avulsion

Stent avulsion is a very rare occurrence facilitated by the ultrathin struts of the third generation, which offer low resistance to the friction of balloons or additional stent passage when the previous stent has been inflated at nominal pressure and slightly malposed. Upon withdrawing, the balloon component can become trapped in the stent struts and forced maneuvers result in stent avulsion, which usually causes no flow or vascular dissection and rupture (Figure 7). The most effective strategy to prevent this, in the authors’ experience, is to inflate the stent at high pressure, being sure to have done a proper POT and to avoid the forced maneuver of withdrawing.

## 4. Conclusions

Technical complications, although rarely occurring and rarely reported in the literature, can be happen during bifurcation PCI, especially in complex bifurcation and with dual-stent techniques. Knowledge of all the different potential complications, the underlying mechanisms, and possible solutions are mandatory for cardiovascular professionals involved in such a challenging field as interventional cardiology.

## Figures and Tables

**Figure 1 jcm-11-06801-f001:**
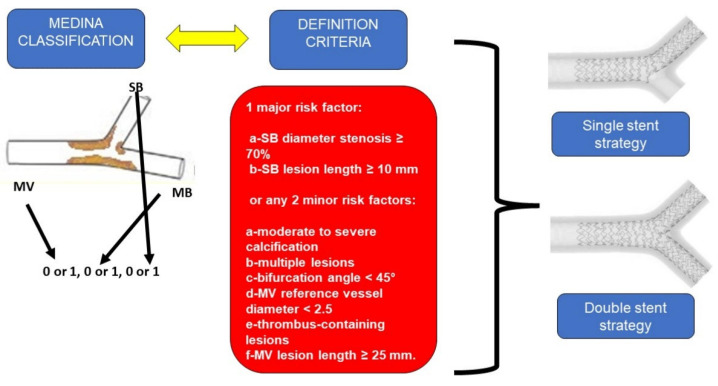
Medina classification and DEFINITION criteria for coronary bifurcation (**left**); general appearance of single stent strategy (**upper right**); general appearance of double stent strategy (**lower right**). SB: Side branch; MV: Main vessel; MB: Main branch.

**Figure 2 jcm-11-06801-f002:**
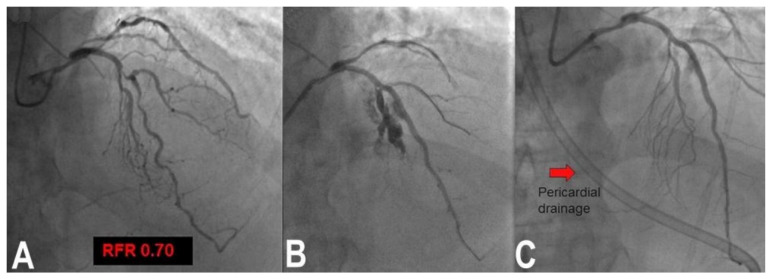
Vessel rupture example in a patient with complex significant FFR bifurcation lesion of left anterior descending and left main coronary artery (**A**). Stent implantation at nominal pressure resulted in vessel rupture and tamponade (**B**). Recovery was obtained by prolonged balloon inflation and autotransfusion of the blood from the pericardial drainage (red arrow, (**C**)).

**Figure 3 jcm-11-06801-f003:**
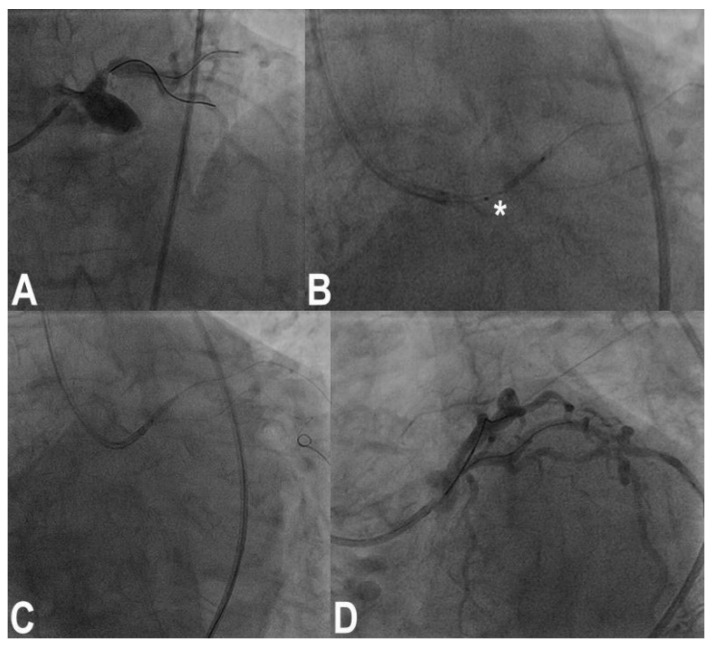
Ostial left anterior descending (LAD) coronary artery treated by crossover stenting from the left main to the LAD (**A**); note the portion of the balloon uncovered by the 4.0 × 13 mm stent during precise positioning of the stent itself (asterisk, (**B**)); original stent balloon was removed and a 1.5 mm balloon was inflated at nominal pressure inside the stent, and then the stent was retracted to the desired position (**C**) with a good final results (**D**).

**Figure 4 jcm-11-06801-f004:**
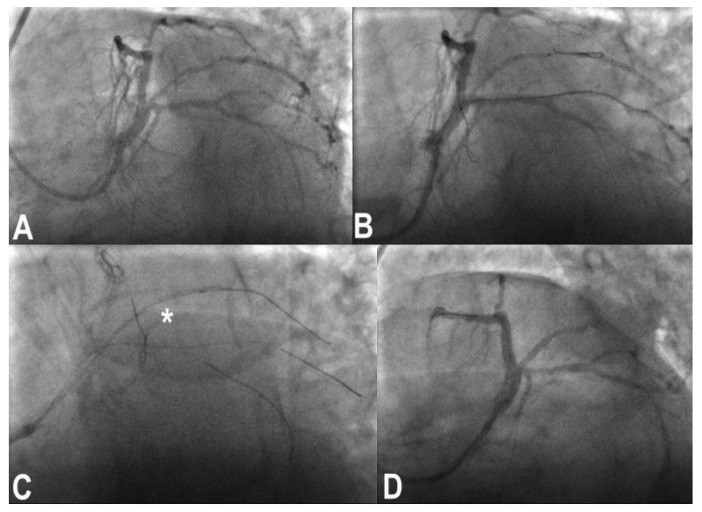
Treatment of a complex trifurcation of left anterior descending coronary artery, ramus and left circumflex artery (**A**). After preparation of the lesion of ramus, a stent was passed with a lot of friction, resulting in stent and wire loss (**B**,**C**) across left main and ramus. A conversion to DK-crush technique was performed with good results (**D**).

**Figure 5 jcm-11-06801-f005:**
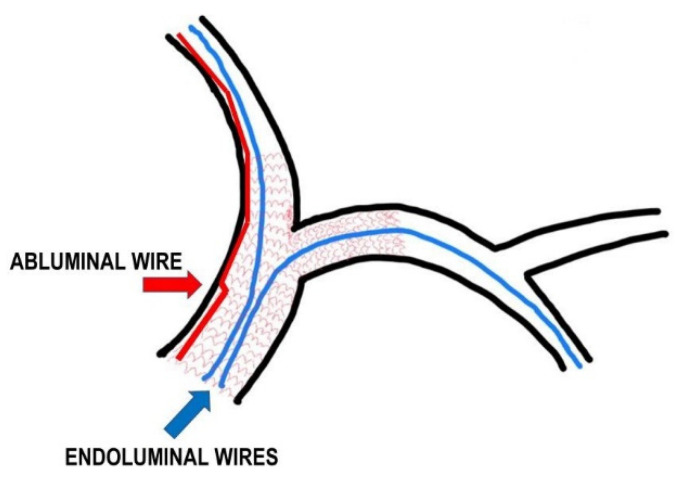
Model of a complex bifurcation treated with the double-stenting strategy. Abluminal wiring: blue line and arrow indicate endoluminal wire crossing, while the red line and arrow indicate a wire which has crossed partially between the artery wall and the stent struts.

**Figure 6 jcm-11-06801-f006:**
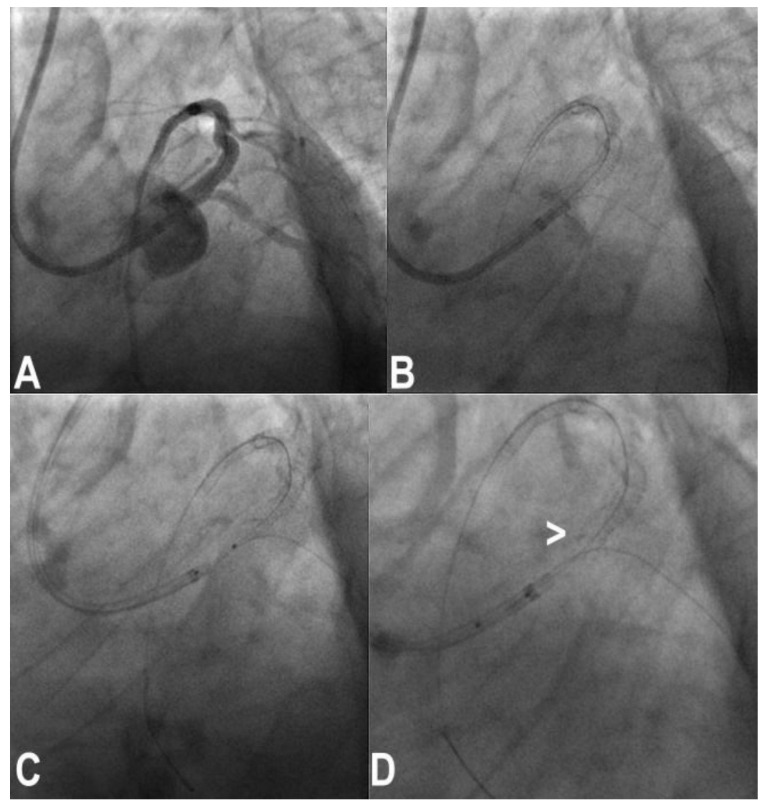
Stent deformation: a stent was placed crossover in left main and left anterior descending (**A**,**B**); an attempt was made to pass a POT balloon through, (**C**) but this resulted in stent deformation and shortening (white arrow, (**D**)).

**Figure 7 jcm-11-06801-f007:**
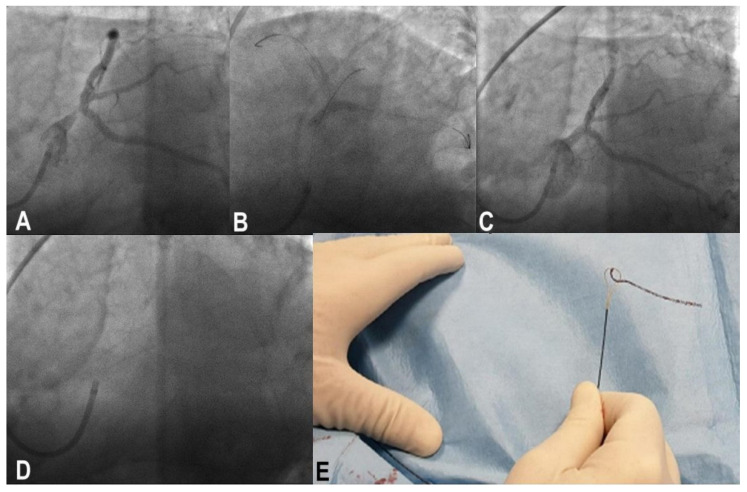
A rare case of stent avulsion in a complex left main (LM) bifurcation disease (**A**) treated by provisional single stent strategy. After stenting of the LM to left anterior descending (LAD) coronary artery, the left circumflex (LCx) was rewired, and a POT-side-PO sequence was performed with the already-used balloon (**B**). After forced withdrawal of the LCx balloon, a no-flow on LAD was apparent, (**C**) and the crossover stent was no longer visible (**D**). (**E**) It was recaptured partially fractured outside the guiding catheter.

**Table 1 jcm-11-06801-t001:** Summary of the stenting techniques in bifurcation, with particular focus on the number of steps in which complications can occur. MB: main branch; MV: main vessel; POT: proximal optimization technique; SB: side branch; DK: Double kissing; TAP: T and protrusion.

Strategy	Technique	Key Features and Steps
Single stent	Provisional (cross-over)	POT, eventually rewiring, Re-POT, eventually kissing balloon step
Double stent		
SB upfront	Culotte	Overlapping of the proximal portion of the MV and SB stent; 1 rewiring; 1 kissing balloon step, 1 POT step
	DK-crush	Crushing of a variable portion of the SB stent by the MV stent; 2 kissing balloon steps, 1 POT step
	Inverted T	SB stent with no protrusion
	Nano-Inverted-T	SB stent with minimal protrusion, 1 rewiring step, 2 POT steps, 1 kissing step
	V-kissing	Simultaneous deployment of MB and SB stents, 1 kissing balloon step
After MV stenting	Culotte	Overlapping of the proximal portion of the MV and SB stent; 1 rewiring; 1 kissing balloon step, 1 POT step
	T-stenting	1 rewiring step, 1 kissing balloon step
	TAP	Slight protrusion of SB stent into the MV stent with neocarena formation, 1 rewiring step, 1 kissing balloon step

**Table 2 jcm-11-06801-t002:** Proposed classification of technical complications during coronary bifurcation percutaneous coronary interventions. LM: left main.

Coronary Segment	Technique	Stage	Technical Complication	Clinical Results (Clinical Complication)
Non-LM bifurcation LM bifurcation	Single stent Double stent	Wire crossing	Dissection Perforation Wire entrapment and fracture	Death Myocardial infarction
		Ballon dilation	Dissection Vessel rupture Balloon rupture Balloon entrapment	Pericardial effusion/Tamponade
		Plaque modification	Vessel rupture Device entrapment	Early stent thrombosis
			Stent loss: -wire in -wire out	
		Stent deployment	Stent underexpansion	
		Wire recrossing	Abluminal recrossing	
		Final Optimization	Balloon rupture Vessel rupture Stent deformation Stent avulsion	

**Table 3 jcm-11-06801-t003:** Specific complications related to different techniques and steps. MV: main vessel; POT: proximal optimization technique; SB: side branch.

Strategy	Technique	Complication	Step
Single stent	Provisional	Dissection	First wiring SB
		Abluminal rewiring	After stent deployment
		Stent deformation	Any stage after stenting
		Stent avulsion	Any stage after stenting
Double stent			
SB upfront	Culotte	Dissection	First wiring SB
			After SB predilation
		Abluminal rewiring	After SB stent deployment
		Wire/balloon entrapment	After SB deployment
		Stent deformation	Any stage after MV stenting
	DK-crush	Dissection	First wiring SB and rewiring
			At first kissing
		Abluminal rewiring	After SB stent deployment and first kissing
		Wire/balloon entrapment	At second kissing
		Stent deformation	Any stage after MV stenting
	Inverted-T and Nano-Inverted-T	Dissection	First wiring SB and rewiring
		Abluminal rewiring	After MV stent deployment and first POT
		Wire/balloon entrapment	At balloon kissing
		Stent deformation	Any stage after MV stenting
	V kissing	Dissection	Wire crossing
			Stent deployment

## Data Availability

Not applicable.

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
