# Peer review of "Technical Complications of Coronary Bifurcation Percutaneous Interventions"

_jcm, 2022, doi:10.3390/jcm11226801_

Round 1

Reviewer 1 Report

The article describes technical complication occurring during coronary bifurcation stenting techniques, providing details very useful in daily clinical practice in the cath lab.

When referring to abluminal wire crossing, this could be a rather frequent issue especially in malaposed stents with ultra thin struts and in complex two stent procedures as DK crush or culotte. These complications may remain unrecognised, especially when happening in side branch stents,  with possible severe outcomes. Therefore,  the results should be checked by intracoronary imaging (OCT providing more details than IVUS due to its superior resolution). Please refer to the article with DOI: 10.4244/EIJ-D-16-00689.

Another stent deformation complication may occur when removing the balloon, elongating stent struts and inducing a severe malaposition (with possible very late stent thrombosis after stopping double antiaggregation)

Please check the Table 1 position in the page (it is not completely visible). Editing the table 2 is also needed.

Author Response

The article describes technical complication occurring during coronary bifurcation stenting techniques, providing details very useful in daily clinical practice in the cath lab. When referring to abluminal wire crossing, this could be a rather frequent issue especially in malapposed stents with ultrathin struts and in complex two stent procedures as DK crush or culotte. These complications may remain unrecognised, especially when happening in side branch stents, with possible severe outcomes. Therefore, the results should be checked by intracoronary imaging (OCT providing more details than IVUS due to its superior resolution). Please refer to the article with DOI: 10.4244/EIJ-D-16-00689.Answer: we added few lines as suggested and we changed the ref with the suggested ref.

Another stent deformation complication may occur when removing the balloon, elongating stent struts and inducing a severe malaposition (with possible very late stent thrombosis after stopping double antiaggregation) Answer: we added this comment to the paragraph

Please check the Table 1 position in the page (it is not completely visible). Editing the table 2 is also needed. Answer: We reupload table I and we re-editing Table 2

Reviewer 2 Report

This review summarizes the procedural complications associated with bifurcation stenting.

Bifurcation stenting is a major PCI controversy and is still under discussion. The authors provide a review of this hot area.

Point by point

1.      Recently, bifurcation techniques have evolved, such as DK-stenting. This review lists complications, but it is difficult to understand the relationship between each technique and complications. A comparison of techniques and their respective complication rates would be useful to caution about complications to the PCI operator.

2.      Although there is a figure of wire recrossing, it might be easier to understand if other complications are shown, especially at which step of the procedure complications are most likely to occur.

Author Response

This review summarizes the procedural complications associated with bifurcation stenting.

Bifurcation stenting is a major PCI controversy and is still under discussion. The authors provide a review of this hot area.

Point by point

  1. Recently, bifurcation techniques have evolved, such as DK-stenting. This review lists complications, but it is difficult to understand the relationship between each technique and complications. A comparison of techniques and their respective complication rates would be useful to caution about complications to the PCI operator. Answer: A comparison of complication rate among the different technique is very difficult to obtain because normally such technical complications are not listed in any series but what are listed are general complication; AMI, stent thrombosis, death. Thus to comply with the reviewer’s suggestion as much as possible, we added a new table listing the technical complications which may occur with each technique and at which step
  2. Although there is a figure of wire recrossing, it might be easier to understand if other complications are shown, especially at which step of the procedure complications are most likely to occur. Answer: we believe that Figure is still informative. We added a new table (see above) to comply with the reviewer’s suggestion